# The socio-economic and health effects of COVID-19 among rural and urban-slum dwellers in Ghana: A mixed methods approach

**Matilda Aberese-Ako**[1]\*, **Mustapha Immurana**[1], **Maxwell Ayindenaba Dalaba**[1], **Fidelis E. Y. Anumu**[1], **Anthony Ofosu**[2], **Margaret Gyapong**[1]

**1** Institute of Health Research, University of Health and Allied Sciences, Ho, Ghana, **2** Ghana Health Service, Accra, Ghana

\* maberese-ako@uhas.edu.gh

## Abstract

### Background

Vulnerable populations such as rural and urban-slum dwellers are more likely to suffer greatly from the deleterious effects of the novel Coronavirus disease 2019 (COVID-19). However, in Ghana, most COVID-19 mitigating packages are not focused on vulnerable populations.

### Methods

Concurrent mixed methods design was used to examine the socio-economic and health effects of COVID-19 among rural and urban-slum dwellers in Ghana. Four hundred respondents were sampled for the quantitative arm of the study, while 46 In-depth Interviews (IDIs) were conducted with community members and government officials. Sixty-four community members participated in Focus Group Discussions (FGDs) and non-participant observation was carried out for three months. Quantitative data were analysed using frequencies, percentages, Pearson Chi2 and ordered logistic regression. Interviews were recorded using digital recorders and later transcribed. Transcribed data (IDIs, FGDs) and observation notes were uploaded onto a computer and transferred to qualitative software NVivo 12 to support thematic coding and analysis.

### Results

Majority of the respondents confirmed the deleterious socio-economic and health effects of COVID-19 on jobs and prices of food. Other effects were fear of visiting a health facility even when unwell, depression and anxiety. Young people (18–32 years), males, urban-slum dwellers, married individuals, the employed and low-income earners (those who earn GHC10/$1.7 to GHC100/ $17), were more likely to suffer from the socio-economic and health effects of COVID-19. Urban-slum dwellers coped by relying on family and social

**Data Availability Statement:** Data are available for researchers who meet the criteria for access to confidential data. They may contact the following

address: Research Ethics Committee, University of Health and Allied Sciences, email: rec@uhas.edu. gh.

**Funding:** MG is the main recipient of funding Financial support for this study was obtained from the World Health Organization [WHO] Alliance for Health Policy and Health Systems Research [2020/ 1037470-0]. The funders had no role in the study design, data collection and analysis, decision to publish, or preparation of the manuscript.

**Competing interests:** The authors have declared that no competing interests exist.

networks for food and other basic necessities, while rural dwellers created locally appropriate washing aids to facilitate hand washing in the rural communities.

## Conclusion

COVID-19 and the government's mitigation measures had negative socio-economic and health effects on vulnerable communities. While vulnerable populations should be targeted for the government's COVID-19 mitigating packages, special attention should be given to young people (18–32 years), males, urban-slum dwellers, married individuals and low-income earners. Communities should be encouraged to maintain coping strategies adopted even after COVID-19.

## Introduction

The alarming pace at which the novel Coronavirus disease 2019 (COVID-19) infected millions of people all over the world, necessitated the imposition of restrictions on movements, social distancing, as well as the use of masks and hand sanitizers in order to contain the spread of the virus. Nonetheless, these restrictions have brought economic activities to a near-standstill, making the pandemic the largest economic shock experienced by the world in decades [1]. Various mitigating packages have been instituted by governments all over the world aimed at cushioning citizens from the damaging effects of COVID-19. It is recommended that, vulnerable populations such as urban-slum and rural dwellers, those in the informal sector, women, children and people with disabilities are given special attention in the mitigation measures [1,2].

The first case of COVID-19 in Ghana was recorded on 12[th] March, 2020. Subsequently, the infection became more concentrated in urban areas in the three most populous cities in the country, Accra, Tema and Kumasi [3]. By June, 2020 the Greater Accra region had recorded 11,348 (63.9%) COVID-19 cases, followed by the Ashanti region with 3,003 (16.9%) COVID-19 cases, while the Volta Region recorded 221 (1.2%) COVID-19 cases [4]. The government of Ghana implemented mitigation measures such as subsidised electricity, free water, GHC 1 billion ($170million) support to small-medium scale firms, extension of tax filing date, reduction in interest rate and a GHC 3 billion ($510 million) support facility for local industries [5]. However, most of these mitigating packages were more formal sector centred. Meanwhile, majority of Ghanaians (71.3%) are employed in the informal sector [6]. Urban-slum and rural dwellers, majority of whom are informal sector actors with no access to piped water and electricity, did not benefit from most of the subsidies [7].

While a number of studies have been conducted on the socio-economic and health impact of COVID-19 in other parts of the world [8–21] as well as in Ghana [22–29], a substantial number of these studies were either conceptual in nature or paid little attention to vulnerable populations.

Among the Ghanaian studies, only the ones by Adom et al. [22] and Saah et al. [29] were solely dedicated to vulnerable people in urban settings. Adom et al. [22] reported that the COVID-19 measures had resulted in economic hardships among the urban poor, while Saah et al. [29] indicated that it had resulted in the fear of seeking health care in public facilities. Other studies reported that the partial lockdown in the Accra and Tema metropolis and the greater Kumasi area, which have the highest concentration of population in Ghana and were the epicentre of infection [3,30], resulted in hardships for the informal economy and the

urban-slum [3]. Some of the effects were loss of jobs, income and lack of food, increase in poverty levels and fear of accessing health care [31–34]. This current study contributes to literature by employing a concurrent triangulation mixed methods design, to examine the socio-economic and health effects of COVID-19 as well as the categories of vulnerable individuals (urban-slum and rural residents) that are more susceptible to the effects of COVID-19 in Ghana. It also reports on the coping strategies of the two populations, making the findings of this study more diversified and relevant for policy and intervention purposes.

## Materials and methods

### Study setting

The study was carried out in the Oforikrom Municipality in the Ashanti region and the Adaklu District of the Volta Region of Ghana.

The Ashanti Region was chosen, because it had the second highest number of COVID-19 cases, yet it had a relatively higher rate of COVID-19 deaths compared to the rest of the country [35]. The Oforikrom Municipality, located in the Ashanti Region, was chosen because it has a large number of urban-slum dwellers who are predominantly internal migrants [36,37]. Further, the municipality was one of the areas that experienced the government's COVID-19 mitigation measures of partial lockdown, so its choice gave this study a perspective of how such an intervention affected urban-slum dwellers. The Municipality is located between latitudes 6.35oN and 6.40oS and longitude 1.30oW and 1.35oE [38].

The Volta Region was chosen, because it is one of the poorest regions in the country. Moreover, 23.1% of households in the Volta Region have drinking water quality problems, while 26.6% have no toilet facilities [6]. The Adaklu District of the Volta Region was chosen because, aside from being found in one of the poorest regions in Ghana [39], it is predominantly rural [40]. The Adaklu District is located on longitudes 06′41′1″N and 6.68361˚N and latitudes 00˚ 20′1″E and 0.33361˚E [41]. The district was carved out of the Adaklu-Anyigbe District (now Agotime Ziope district), which had the highest rural population in the Volta Region, per the 2010 population and housing census in Ghana [40]. Thus, choosing the Adaklu District sheds light on perspectives of rural communities with regard to COVID-19.

### Study design and population

The study adopted a concurrent triangulation mixed methods design to obtain different but complementary data on the same topic [42,43]. Quantitative and qualitative data were collected concurrently, from October to December, 2020, by two separate team of researchers. The quantitative team administered survey questionnaires to 400 participants, while the qualitative team used focused ethnography that employed in-depth interviews (IDIs), focus group discussions (FGDs) and non-participant observation. The two datasets were triangulated in the interpretation of the results and discussion sections of the study. The 'Good Reporting of A Mixed Methods Study (GRAMMS)' checklist [44] was used in reporting the study and has been included as supporting information.

### Quantitative data collection process and analysis

**Sample size and sampling procedure.** The quantitative aspect employed a four-stage sampling technique to select respondents. In the first stage, the Ashanti and the Volta Regions were purposively selected. In the second stage, the Oforikrom Municipality in the Ashanti Region and the Adaklu District in the Volta Region were selected purposively. In the third stage, purposive sampling was used to select two urban-slum communities from the

Oforikrom Municipality and two rural communities from the Adaklu District. Using the Krejcie and Morgan's formula [45], we arrived at a sample size of 384, which was adjusted for possible non response to 400.

In the fourth stage, we used the simple random sampling technique to select 200 respondents from each of the study sites. Respondents were males and females who were 18 years and above. The random sampling used ensured that each respondent had an equal chance of being selected.

**Data collection, quality control and assurance.** The quantitative data was collected with the aid of a structured questionnaire prepared on mobile tablets with Redcap (which captured information about the socio-economic and demographic background of respondents as well as the socio-economic and health effects of COVID-19). As a form of quality control, prior to the final data collection, the questionnaire was pretested in communities with similar features as the chosen communities and all necessary corrections were made. Final data collection was done by eight graduate level Research Assistants (RAs: four in each district/municipality), who were trained for three days. As a further quality control check, research assistants were supervised throughout the data collection process, to ensure that they adhered to all ethical and scientific procedures. At the time of data collection (October to December, 2020), restrictions on movements had been relaxed in Ghana, which paved the way for in-person data collection.

**Data management and analysis.** After the data collection, the data was downloaded from Redcap in the form of a STATA file to facilitate analysis using STATA 14.0. The negative effects of COVID-19 on jobs, rising prices of food items and the inability to afford enough food were used as indicators for the socio-economic effects; while fear to seek healthcare in a health facility even when unwell, depression and anxiety were used as indicators for the health effects. Depression and anxiety that were used in this study were based on respondents' perceptions. The socio-economic and health variables were measured as: 1 (strongly disagree), 2 (disagree), 3 (agree) and 4 (strongly agree).

Frequencies and percentages were used to assess the extent to which respondents agreed to the effects of COVID-19 on the above-mentioned indicators. The Pearson Chi2 technique was used to assess if there were significant differences between males and females, as well as urban-slum and rural dwellers with regard to the socio-economic and health effects of COVID-19. Given the ordinal nature of the dependent variables (the socio-economic and health effects named-above), we employed the ordered logistic regression method [46], to explore factors that influence susceptibility to the effects of COVID-19. In running the regressions, age, sex, residence type, marital status, income, religion, education and employment status were used as independent variables. The independent variables were treated as dummy variables due to their categorical nature.

## Qualitative data collection process and analysis

**Characteristics of study participants and sampling procedure.** To establish rigour and credibility of the qualitative data, triangulation was employed by using three data collection methods IDIs, FGDs and non-participant observation [47]. Forty-six IDIs were conducted with chiefs, elders, persons with disabilities, persons who had tested positive for COVID-19 or whose relatives had tested positive and community healthcare volunteers (healthcare volunteers in this study refer to community members who support healthcare workers in Community-Based Health Planning and Services (CHPS) compounds and health centres in service provision), in the two study areas. In addition, IDIs were conducted with healthcare providers from the Ghana Health Service and desk officers responsible for COVID-19 relief items in the two study districts. A total of 64 people comprising 32 males and 32 females participated in eight FGDs; 4 FGDs in each of the study sites (Table 2).

To understand the multiple layers of the effect of COVID-19 and the government's mitigation measures in a natural setting and to prevent bias, a representative purposive sampling technique (maximum variation sampling) [48–50], was used to select government officials and community leaders in the two study areas. Convenience sampling was used to select community members to participate in FGDs. Data saturation was attained when no new information on the study themes was obtained from the different sampled groups, which is in line with Charmaz's recommendation [51].

**Qualitative data collection process and quality control and assurance.** Eight graduate RAs were trained by a medical anthropologist (MA) to conduct the qualitative interviews and to observe. Four RAs were assigned to each of the two study sites. To ensure quality control, the study guides (IDI and FGD) were pre-tested in two communities with similar characteristics as the study communities. Additionally, two PhD level officers supervised the qualitative data collection and held regular meetings with the RAs to guide the data collection process. Six of the audio interviews were transcribed by two RAs as part of the quality control process and also to ensure rigour in the quality of data collected.

IDIs and FGDs sought to understand the socio-economic effects of the pandemic, effect on health seeking behaviour and coping strategies among others. Interviews were conducted in the predominant local languages of the area (Hausa and Dagbani) for those in the Ashanti Region and Ewe for those in the Volta Region. All government officials such as health workers and staff of the Ministry of Gender were interviewed in English, as English is the official language of Ghana.

A checklist was used to observe daily activities and interactions in study communities, availability of COVID-19 prevention items in homes and public spaces, and the observance of COVID-19 prevention protocols. In addition, smart phones were used to take pictures of some of the critical observations such as social gatherings, as well as the location of COVID-19 prevention items in public places and in households. The IDIs took a period of 30 to 35 minutes, while the FGDs lasted an average of one hour and 20 minutes to conduct.

**Data management and analysis.** Interviews were recorded digitally and were translated and transcribed verbatim into English to maintain responses from study participants. Transcribed IDIs, FGDs and observation notes were uploaded onto a computer and transferred to qualitative software Nvivo 12 to support data coding and analysis. The data was triangulated for analysis. A code book was developed based on the study themes and it facilitated thematic analysis, which formed the basis for reporting study results.

## Ethics statement

Ethical approval for this study was obtained in 2020, from the University of Health and Allied Sciences' Research Ethics Committee (UAHS-REC A.2 [8] 20–21), Ho Ghana. Ethical procedures were followed (voluntary consent of all study participants was sought, they were assured of the voluntariness of the study and confidentiality of their identity).

The study offered one face mask to each of the study participants, thus, both the RAs and the respondents wore face masks, while observing social distancing throughout the interview period. Also, the RAs sanitised their hands and that of the respondents, prior to the interview, while the mobile tablets were sanitised after each interview using an alcohol-based sanitiser.

## Results

### Socio-economic background of respondents

The background characteristics of study participants in the quantitative study and the category of persons who participated in the qualitative study are presented in Tables 1 and 2 respectively.

**Table 1. Socio-economic background of respondents.**

| Variable | Freq. | Percent |
|---|---|---|
| **Sex** | | |
| Male | 197 | 49.25 |
| Female | 203 | 50.75 |
| Total | 400 | 100.00 |
| **Age** | | |
| 18–32 years | 202 | 50.63 |
| 33–46 years | 102 | 25.56 |
| 47–60 years | 64 | 16.04 |
| 61 years and above | 31 | 7.77 |
| Total | 399 | 100.00 |
| **Residence** | | |
| Rural | 200 | 50.00 |
| Urban-slum | 200 | 50.00 |
| Total | 400 | 100.00 |
| **Marital status** | | |
| Married | 210 | 52.50 |
| Single | 128 | 32.00 |
| Divorced | 13 | 3.25 |
| Widowed | 21 | 5.25 |
| Separated | 8 | 2.00 |
| Cohabitating | 20 | 5.00 |
| Total | 400 | 100.00 |
| **Religion** | | |
| Christianity | 198 | 49.50 |
| Islam | 196 | 49.00 |
| Traditionalist | 5 | 1.25 |
| None | 1 | 0.25 |
| Total | 400 | 100.00 |
| **Education** | | |
| Basic | 209 | 53.05 |
| Secondary | 64 | 16.24 |
| Tertiary | 14 | 3.55 |
| None | 107 | 27.16 |
| Total | 394 | 100.00 |
| **Income** | | |
| GHC10-GHC100 | 102 | 25.69 |
| GHC101-GHC500 | 163 | 41.06 |
| GHC501-GHC1000 | 91 | 22.92 |
| GHC1001 and above | 41 | 10.33 |
| Total | 397 | 100.00 |
| **Employment** | | |
| Employed | 293 | 73.25 |
| Unemployed | 107 | 26.75 |
| Total | 400 | 100.00 |

Observations less than 400 represent cases of missing data.

**Table 2. Qualitative data collection methods and category of study respondents.**

| Category of respondents who participated in IDIs | Study Areas | |
|---|---|---|
| | Oforikrom Municipality | Adaklu District |
| **Category** | **No.** | **No.** |
| Assembly members | 1 | 1 |
| Chiefs and queen mothers | 2 | 2 |
| Community elders | 2 | 2 |
| Staff of the Ministry of Gender and Social Protection | 1 | 1 |
| Ghana Health Service officials | 2 | 2 |
| Religious leaders | 2 | 2 |
| Herbalists | 2 | 2 |
| Persons who have been affected directly or indirectly by COVID-19 | 3 | 3 |
| Persons with disability (case studies) | 3 | 3 |
| Persons caring for persons with disabilities (case studies) | 3 | 3 |
| Community volunteers | 2 | 2 |
| **Total number of interviews** | **23** | **23** |
| **Category of respondents who participated in FGDs** | | |
| Men under 30 years) | 1 | 1 |
| Women under 30years | 1 | 1 |
| Men over 30 years | 1 | 1 |
| Women over 30years | 1 | 1 |
| **Total number of FGDs** | **4** | **4** |

*Non-Participant observation was conducted alongside the collection of IDIs and FGDs.

A little over half of the respondents were females (50.75%) and 50.63% were in the youngest age category of 18–32 years. About 53.05% of respondents had basic level of education, while 16.24% and 3.55% had secondary and tertiary levels of education respectively. While most of the respondents were employed (73.25%), only few of them (10.33%) earned a monthly income of GHC1001 ($170.17) and above (Table 1).

## Socio-economic and health effects of COVID-19

Majority of the respondents (93.86%) agreed that COVID-19 had negatively impacted their jobs, and 89.75% of the respondents stated that COVID-19 had made food items expensive (rising prices) and 76.25% said they were unable to afford enough food, because of the negative effects of COVID-19 (Table 3).

Concerning health seeking behaviour, 65.50% of the respondents in the quantitative study confirmed their fear to seek healthcare in a health facility even if they were unwell. Similarly, 74.75% and 70.75% of the respondents respectively, stated that they felt anxious and depressed due to COVID-19 (Table 3).

Participants in both the urban-slum and rural communities reported that the effects of COVID-19 and the public health measures imposed by the government made it difficult for them to continue with their businesses, as most of them lost their sources of livelihood.

Majority of participants in the rural setting were either farmers or owned farms as a second job. They had access to their farms, because the state's restrictions in the rural community were less stringent compared to the urban-slum, besides the nature of farming does not require much interactions. However, they complained of their inability to hire farm labour, so they could not farm large plots of land. To make matters worse, some of their products got rotten

Table 3. Socio-economic and health effects of COVID-19.

| Variable | Frequency | Percent |
|---|---|---|
| **Negative effect on job** | | |
| Strongly disagree | 11 | 3.75 |
| Disagree | 7 | 2.39 |
| Agree | 46 | 15.70 |
| Strongly Agree | 229 | 78.16 |
| Total | 293[a] | 100.00 |
| **Prices of food items have increased** | | |
| Strongly disagree | 10 | 2.50 |
| Disagree | 31 | 7.75 |
| Agree | 160 | 40.00 |
| Strongly Agree | 199 | 49.75 |
| Total | 400 | 100.00 |
| **Inability to afford enough food** | | |
| Strongly disagree | 15 | 3.75 |
| Disagree | 80 | 20.00 |
| Agree | 189 | 47.25 |
| Strongly Agree | 116 | 29.00 |
| Total | 400 | 100.00 |
| **Afraid to seek healthcare even if unwell** | | |
| Strongly disagree | 72 | 18.00 |
| Disagree | 66 | 16.50 |
| Agree | 123 | 30.75 |
| Strongly Agree | 139 | 34.75 |
| Total | 400 | 100.00 |
| **Feel depressed** | | |
| Strongly disagree | 50 | 12.50 |
| Disagree | 67 | 16.75 |
| Agree | 179 | 44.75 |
| Strongly Agree | 104 | 26.00 |
| Total | 400 | 100.00 |
| **Feel anxious** | | |
| Strongly disagree | 50 | 12.50 |
| Disagree | 51 | 12.75 |
| Agree | 179 | 44.75 |
| Strongly Agree | 120 | 30.00 |
| Total | 400 | 100.00 |

[a] The question on the negative effect on job was restricted to only those who were employed.

on the farms. Those who were able to harvest their products could not sell them because, markets were closed for some time. A large number of study participants said that they had depleted their savings due to their inability to work.

> "*Going to the farm. . . was not a problem for me. I feel like when I go to farm, I am not going to meet anyone. . . The problem rather was about the crops that we have been sending to the market. This is because at that time, they closed some markets. Selling became difficult in those days. . . . Everything got spoilt*!" (IDI, Rural resident, healthcare volunteer01)

Study participants in the urban-slum reported that the nature of the lockdown in their community was extreme. They were restricted to their homes and could not go out to work and carry out other basic necessities of life. This was because, the soldiers detailed to guard their communities to enforce the lockdown, beat them up, whenever they attempted to leave their communities.

> *"When COVID-19 came, we suffered a lot. This suffering was all about how we were going to eat, drink and bath. It brought a whole lot of difficulties to us. We suffered a lot and in addition, we were beaten by soldiers. When you come out to find something to eat and drink, they will beat you and ask you to stay in the rooms. How can you stay in your room [rooms are poorly ventilated and do not contain basic amenities] and be eating and drinking? They worried us a lot."* (FGD, Urban-slum residents, men under 30 years)

Those in the urban-slum lost both access to physical cash (as they were not able to work) and food to feed their families (since they did not have farms). Some sold out their goods cheaply in order to be able to have money to live on, while others lost their customer base. A study participant shared his experience:

> "*There is no money these days, because my customers who are supposed to come and buy my meat* [butcher by trade], *are not coming due to the challenges that they are also facing in their work. Many schools that buy meat and sell it to their students have been closed.*" (IDI, Urban-slum resident, COVID-19 affected01).

Study participants reported that they were afraid to visit health facilities to seek healthcare, because they perceived that they could be stigmatised, tested for COVID-19 or become infected with COVID-19. Study participants, especially in the rural study community held the misconception that coughing even without a COVID-19 test, could be attributed to COVID-19, so it was best to avoid visiting healthcare facilities, if one had a cough. Such misconception and fear contributed to reduction and delay in the utilisation of healthcare services.

Statistically significant differences were found among males and females ($\chi^2 = 6.32$, p = 0.097), concerning inability to afford enough food, because of the negative effects of COVID-19 (Table 4).

There were significant differences concerning feeling depressed as a result of COVID-19 among males and females ($\chi^2 = 9.67$, p = 0.022). More males (38.07%) strongly agreed that they felt anxious of COVID-19 relative to females (22.17) ($\chi^2 = 12.30$, p = 0.006) (Table 4).

In the urban-slum, some community members lost relatives who were diagnosed with COVID-19 in public hospitals. However, the hospitals did not conduct any tests on the surviving relatives and no contact tracing was done. The relatives reported that none of them got ill. Such experiences contributed to high community distrust of the healthcare system and the government, so community members became reluctant to access healthcare services. Such experiences resulted in an increase in self-medication, use of herbs such as the azadirachta indica plant [neem] and the hibiscus plant to treat ailments. Study participants shared experiences as follows:

> "*I hardly go to the hospital like before, but I buy medicine from people who sell it.*" (FGD, Rural residents, women below 30 years).

> "*It [COVID-19] has really influenced my healthcare seeking behaviour because, I have stopped going to the hospital. I have an eye problem, so I often go to hospital for medical*

**Table 4. Socio-economic and health effects of COVID-19, by sex.**

| Sex | Negative effect on job | | | | |
|---|---|---|---|---|---|
| | Strongly disagree | Disagree | Agree | Strongly Agree | Total |
| Male | 4 | 4 | 21 | 119 | 148 |
| | 2.70 | 2.70 | 14.19 | 80.41 | 100.00 |
| Female | 7 | 3 | 25 | 110 | 145 |
| | 4.83 | 2.07 | 17.24 | 75.86 | 100.00 |
| Total | 11 | 7 | 46 | 229 | 293 |
| | 3.75 | 2.39 | 15.70 | 78.16 | 100.00 |

Pearson Chi2 = 1.63 Prob = 0.6521

| | Prices of food items have increased | | | | |
|---|---|---|---|---|---|
| | Strongly disagree | Disagree | Agree | Strongly Agree | Total |
| Male | 4 | 16 | 72 | 105 | 197 |
| | 2.03 | 8.12 | 36.55 | 53.30 | 100.00 |
| Female | 6 | 15 | 88 | 94 | 203 |
| | 2.96 | 7.39 | 43.35 | 46.31 | 100.00 |
| Total | 10 | 31 | 160 | 199 | 400 |
| | 2.50 | 7.75 | 40.00 | 49.75 | 100.00 |

Pearson Chi2 = 2.55 Prob = 0.4662

| | Unable to afford enough food | | | | |
|---|---|---|---|---|---|
| | Strongly disagree | Disagree | Agree | Strongly Agree | Total |
| Male | 7 | 44 | 81 | 65 | 197 |
| | 3.55 | 22.34 | 41.12 | 32.99 | 100.00 |
| Female | 8 | 36 | 108 | 51 | 203 |
| | 3.94 | 17.73 | 53.20 | 25.12 | 100.00 |
| Total | 15 | 80 | 189 | 116 | 400 |
| | 3.75 | 20.00 | 47.25 | 29.00 | 100.00 |

Pearson Chi2 = 6.32 Prob = 0.0968

| | Afraid to seek healthcare even if unwell | | | | |
|---|---|---|---|---|---|
| | Strongly disagree | Disagree | Agree | Strongly Agree | Total |
| Male | 37 | 34 | 56 | 70 | 197 |
| | 18.78 | 17.26 | 28.43 | 35.53 | 100.00 |
| Female | 35 | 32 | 67 | 69 | 203 |
| | 17.24 | 15.76 | 33.00 | 33.99 | 100.00 |
| Total | 72 | 66 | 123 | 139 | 400 |
| | 18.00 | 16.50 | 30.75 | 34.75 | 100.00 |

Pearson Chi2 = 1.02 Prob = 0.7971

| | Feel depressed | | | | |
|---|---|---|---|---|---|
| | Strongly disagree | Disagree | Agree | Strongly Agree | Total |
| Male | 18 | 36 | 81 | 62 | 197 |
| | 9.14 | 18.27 | 41.12 | 31.47 | 100.00 |
| Female | 32 | 31 | 98 | 42 | 203 |
| | 15.76 | 15.27 | 48.28 | 20.69 | 100.00 |
| Total | 50 | 67 | 179 | 104 | 400 |
| | 12.50 | 16.75 | 44.75 | 26.00 | 100.00 |

Pearson Chi2 = 9.67 Prob = 0.0216

| | Feel anxious | | | | |
|---|---|---|---|---|---|
| | Strongly disagree | Disagree | Agree | Strongly Agree | Total |
| Male | 21 | 21 | 80 | 75 | 197 |

(*Continued*)

**Table 4.** (Continued)

|  | | | | | |
|---|---|---|---|---|---|
|  | 10.66 | 10.66 | 40.61 | 38.07 | 100.00 |
| Female | 29 | 30 | 99 | 45 | 203 |
|  | 14.29 | 14.78 | 48.77 | 22.17 | 100.00 |
| Total | 50 | 51 | 179 | 120 | 400 |
|  | 12.50 | 12.75 | 44.75 | 30.00 | 100.00 |

Pearson Chi2 = 12.30 Prob = 0.0064

First row has *frequencies* and second row has *row percentages*.

*check-up, but due to what I have gone through [grandmother died of COVID-19 in the hospital], I have stopped going to the hospital for medical check-up."* (IDI, Urban-slum resident, COVID-19 affected02)

A total of 71.63% and 84.21% of rural and urban-slum dwellers respectively, strongly agreed that COVID-19 had impacted their jobs negatively. The relationship between residence type and the deleterious effect of COVID-19 on jobs was statistically significant ($\chi^2$ = 12.41, p = 0.006) (Table 5).

There was a statistically significant ($\chi^2$ = 8.46, p = 0.037) association between residence type and rising prices of food items, due to COVID-19. Relative to rural dwellers (44%), most residents of the urban-slum (55.50%) strongly agreed that COVID-19 had led to an increase in prices of food items. Concerning inability to afford enough food as a repercussion of COVID-19, we found statistically significant differences among residence types ($\chi^2$ = 12.57, p = 0.006). Specifically, compared to rural dwellers (23.50%), 34.50% of urban-slum residents strongly agreed that COVID-19 had negatively affected their ability to afford enough food.

There were statistically significant differences among urban-slum and rural dwellers with regard to being afraid to seek healthcare even if unwell ($\chi^2$ = 16, p = 0.001). Thus, 41.50% of urban-slum residents strongly agreed that they were afraid to seek healthcare from a health facility even if they were unwell, because of COVID-19, relative to 28% of rural residents.

Statistically significant differences were also found among the two resident types with regard to feeling depressed ($\chi^2$ = 44.80, p = 0.000) and anxious ($\chi^2$ = 51.64, p = 0.000), due to COVID-19. Relative to rural dwellers (19%), most urban-slum dwellers (33%) strongly agreed that they were depressed due to COVID-19. In addition, 41% of urban-slum dwellers strongly agreed that they felt anxious because of COVID-19 relative to 19% of rural dwellers.

## Determinants of susceptibility to the socio-economic and health effects of COVID-19

Respondents aged 61 years and above had 0.26 lower odds of experiencing rising prices of food items as a result of COVID-19 compared to those aged 18 to 32 years (at 1% level of significance). Similarly, respondents aged 47–60 years and those aged 61 years and above, had 0.56 times and 0.20 times lower odds of being unable to afford enough food due to COVID-19 compared to those who were aged 18 to 32 years, at 10% and 1% levels of significance, respectively (Table 6).

Respondents who were single were found to have 0.43 times lower odds of being unable to afford enough food due to COVID-19, relative to their married counterparts at 1% level of significance (Table 6).

**Table 5. Socio-economic and health effects of COVID-19, by residence type.**

| Residence | Negative effect on job | | | | |
|---|---|---|---|---|---|
| | Strongly disagree | Disagree | Agree | Strongly Agree | Total |
| Rural | 8 | 7 | 25 | 101 | 141 |
| | 5.67 | 4.96 | 17.73 | 71.63 | 100.00 |
| Urban-slum | 3 | 0 | 21 | 128 | 152 |
| | 1.97 | 0.00 | 13.82 | 84.21 | 100.00 |
| Total | 11 | 7 | 46 | 229 | 293 |
| | 3.75 | 2.39 | 15.70 | 78.16 | 100.00 |

Pearson Chi2 = 12.41 Prob = 0.0061

| | Prices of food items have increased | | | | |
|---|---|---|---|---|---|
| | Strongly disagree | Disagree | Agree | Strongly Agree | Total |
| Rural | 8 | 19 | 85 | 88 | 200 |
| | 4.00 | 9.50 | 42.50 | 44.00 | 100.00 |
| Urban-slum | 2 | 12 | 75 | 111 | 200 |
| | 1.00 | 6.00 | 37.50 | 55.50 | 100.00 |
| Total | 10 | 31 | 160 | 199 | 400 |
| | 2.50 | 7.75 | 40.00 | 49.75 | 100.00 |

Pearson Chi2 = 8.46 Prob = 0.0373

| | Unable to afford enough food | | | | |
|---|---|---|---|---|---|
| | Strongly disagree | Disagree | Agree | Strongly Agree | Total |
| Rural | 11 | 50 | 92 | 47 | 200 |
| | 5.50 | 25.00 | 46.00 | 23.50 | 100.00 |
| Urban-slum | 4 | 30 | 97 | 69 | 200 |
| | 2.00 | 15.00 | 48.50 | 34.50 | 100.00 |
| Total | 15 | 80 | 189 | 116 | 400 |
| | 3.75 | 20.00 | 47.25 | 29.00 | 100.00 |

Pearson Chi2 = 12.57 Prob = 0.0057

| | Afraid to seek healthcare even if unwell | | | | |
|---|---|---|---|---|---|
| | Strongly disagree | Disagree | Agree | Strongly Agree | Total |
| Rural | 49 | 37 | 58 | 56 | 200 |
| | 24.50 | 18.50 | 29.00 | 28.00 | 100.00 |
| Urban-slum | 23 | 29 | 65 | 83 | 200 |
| | 11.50 | 14.50 | 32.50 | 41.50 | 100.00 |
| Total | 72 | 66 | 123 | 139 | 400 |
| | 18.00 | 16.50 | 30.75 | 34.75 | 100.00 |

Pearson Chi2 = 16.00 Prob = 0.0011

| | Feel depressed | | | | |
|---|---|---|---|---|---|
| | Strongly disagree | Disagree | Agree | Strongly Agree | Total |
| Rural | 44 | 42 | 76 | 38 | 200 |
| | 22.00 | 21.00 | 38.00 | 19.00 | 100.00 |
| Urban-slum | 6 | 25 | 103 | 66 | 200 |
| | 3.00 | 12.50 | 51.50 | 33.00 | 100.00 |
| Total | 50 | 67 | 179 | 104 | 400 |
| | 12.50 | 16.75 | 44.75 | 26.00 | 100.00 |

Pearson Chi2 = 44.80 Prob = 0.0000

| | Feel anxious | | | | |
|---|---|---|---|---|---|
| | Strongly disagree | Disagree | Agree | Strongly Agree | Total |
| Rural | 43 | 36 | 83 | 38 | 200 |

*(Continued)*

**Table 5.** (Continued)

|  |  | 21.50 | 18.00 | 41.50 | 19.00 | 100.00 |
|---|---|---|---|---|---|---|
| Urban-slum |  | 7 | 15 | 96 | 82 | 200 |
|  |  | 3.50 | 7.50 | 48.00 | 41.00 | 100.00 |
| Total |  | 50 | 51 | 179 | 120 | 400 |
|  |  | 12.50 | 12.75 | 44.75 | 30.00 | 100.00 |

Pearson Chi2 = 51.64 Prob = 0.0000

First row has *frequencies* and second row has *row percentages*.

In Table 7, with regard to the health effects of COVID-19, we found that respondents who were 61 years and above had 0.45 times lower odds of feeling depressed due to COVID-19, relative to those who were 18 to 32 years of age (at 10% level of significance). Also, respondents aged 33 to 46 years were found to have 0.58 times lower odds of feeling anxious due to COVID-19 relative to those aged 18 to 32 years, at 5% level of significance.

The findings further show that, urban-slum dwellers had 3.75 times higher odds (significant at 10%) of feeling anxious due to COVID-19 relative to their counterparts in the rural areas. We found that female respondents had 0.62 times (significant at 5%) and 0.57 times (significant at 1%) lower odds of feeling depressed and anxious due to COVID-19 respectively, relative to their male counterparts.

Relative to married respondents, those who were single had 0.54 times and 0.43 times lower odds of feeling depressed and anxious, due to COVID-19, at 5% and 1% levels of significance respectively.

Respondents who were divorced were found to be less likely not to seek healthcare from a health facility if they were unwell, relative to their married counterparts. Specifically, respondents who were divorced had 0.35 times lower odds of not seeking healthcare from a health facility even if unwell, as compared to those who were married (at 5% level of significance). In addition, compared to respondents who were married, those who were cohabiting had 0.22 times and 0.41 times lower odds of feeling depressed and anxious due to COVID-19, at 1% and 5% levels of significance respectively.

Study participants who were employed had 1.83 times higher odds of feeling anxious about COVID-19 relative to those who were unemployed (at 5% level of significance). Notwithstanding, regarding income, the findings show that, those who earned GHC1001 ($170.17) and above, and GHC501 ($85.17) to GHC1000 ($170) a month, had 0.32 times (at 1% level of significance) and 0.48 times (at 5% level of significance) lower odds of not vising a health facility to seek healthcare, even if unwell as well as feeling anxious about COVID-19, respectively, relative to those who earned between GHC10 ($1.7) and GHC100 ($17) a month.

Study participants in the rural community reported in IDIs and FGDs that, one of the negative effects of COVID-19 was that community members were no longer benefiting from public programmes.

Participants who were in tertiary institutions reported that they could not earn an income to pay school fees.

*"To be honest with you, things were very hard when corona came. Some of us pay our own school fees. . .Up till now, I have not yet paid my school fees, because at the time we were supposed to do some work and get money, we could not."* (IDI, Rural resident, Community volunteer01).

**Table 6. Determinants of susceptibility to the socio-economic effects of COVID-19.**

| | Negative effect on job[#] | Increased prices of food items | Unable to afford enough food |
|---|---|---|---|
| **Age (Ref: 18–32 years)** | | | |
| 33–46 years | 1.437 | 0.794 | 0.701 |
| | (0.579) | (0.225) | (0.194) |
| 47–60 years | 1.592 | 0.739 | 0.564* |
| | (0.743) | (0.244) | (0.179) |
| 61 years and above | 0.673 | 0.259*** | 0.197*** |
| | (0.528) | (0.125) | (0.0940) |
| **Residence (Ref: Rural)** | | | |
| Urban-slum | 1.676 | 2.398 | 0.724 |
| | (1.784) | (1.745) | (0.501) |
| **Sex (Ref: Male)** | | | |
| Female | 0.846 | 0.832 | 0.838 |
| | (0.279) | (0.182) | (0.179) |
| **Religion (Ref: Christianity)** | | | |
| Islam | 1.166 | 0.566 | 2.537 |
| | (1.244) | (0.415) | (1.761) |
| Traditionalist | 1.024 | 4.290 | 1.165 |
| | (1.211) | (4.931) | (1.156) |
| None | 0.0754 | 0.270 | 0.799 |
| | (0.124) | (0.461) | (1.354) |
| **Marital status (Ref: Married)** | | | |
| Single | 0.715 | 0.827 | 0.431*** |
| | (0.267) | (0.223) | (0.113) |
| Divorced | 0.165*** | 0.979 | 0.693 |
| | (0.114) | (0.522) | (0.371) |
| Widowed | 0.567 | 1.000 | 1.091 |
| | (0.410) | (0.527) | (0.540) |
| Separated | 1.236 | 0.476 | 1.190 |
| | (1.405) | (0.318) | (0.791) |
| Cohabitating | 0.483 | 0.657 | 0.807 |
| | (0.317) | (0.316) | (0.368) |
| **Education (Ref: None)** | | | |
| Basic | 0.542 | 0.823 | 1.125 |
| | (0.229) | (0.219) | (0.282) |
| Secondary | 0.839 | 1.232 | 1.705 |
| | (0.491) | (0.438) | (0.579) |
| Tertiary | 0.740 | 0.735 | 1.136 |
| | (0.727) | (0.430) | (0.707) |
| **Income (Ref: GHC10-GHC100)** | | | |
| GHC101-GHC500 | 0.978 | 1.202 | 1.176 |
| | (0.414) | (0.323) | (0.307) |
| GHC501-GHC1000 | 1.317 | 1.366 | 1.117 |
| | (0.662) | (0.450) | (0.365) |
| GHC1001 and above | 0.832 | 0.757 | 0.529 |
| | (0.483) | (0.309) | (0.214) |
| **Employment (Ref: Unemployed)** | | | |
| Employed | | 1.196 | 1.248 |

*(Continued)*

**Table 6.** (Continued)

| | Negative effect on job# | Increased prices of food items | Unable to afford enough food |
|---|---|---|---|
| | | (0.315) | (0.320) |
| Observations | 286 | 390 | 390 |
| Chi2 | 23.68 | 29.51 | 48.28 |
| P-value | 0.209 | 0.0782 | 0.000389 |

Exponentiated coefficients (odds ratios) were used; Standard errors in parentheses; * $p < 0.1$, ** $p < 0.05$, *** $p < 0.01$; #: Only one variable in the negative effect on job results is significant, hence, it is not surprising that the overall p-value of the regression is insignificant. We therefore don't make conclusions using regression results in column 2; Observations for the negative effect on jobs model, and increased prices of food as well as inability to afford enough food models, were not up to 293 and 400 respectively because of missing data.

Other effects of COVID-19 on rural dwellers were it reduced community members' desire to support one another and to participate in communal activities. A study participant in an FGD stated: *"Initially, we run to our neighbours in time of need. . ., but with the emergence of COVID-19, it is very difficult to approach your neighbour for help."* (FGD, Rural residents, men below 30years).

Interviews with urban-slum residents and observations revealed that urban-slum dwellers experienced more challenges compared to their rural counterparts, because of the extreme nature of the lockdown and an existing problem of lack of publicly funded amenities such as water, toilets, hospitals and housing in the urban area (Urban-slum, observation notes, 23/11/ 2020). Thus, COVID-19 and the mitigating measures exacerbated urban-slum residents' inability to pay for basic necessities such as food to feed their families, clothing and school fees in quality private basic schools. A butcher who used to slaughter two cows for sale in one day, reported that currently, he is not able to sell the meat of one cow in one day. So, he loses 500 GHS ($85) to 600 GHS ($102) per cow, which has impacted negatively on the standard of living of his family. He shared his experience:

*"There are a lot of changes. Before the advent of COVID-19, when you see the kind of footwear my son used to wear, you will know that he was enjoying, but currently, even my children know I have no money. They used to attend elite schools and not these cheap schools [with-drew children from a top-quality basic school to a lower quality, but cheaper school]"* (IDI, Urban-slum resident, COVID-19 affected01)

The urban-slum community dwellers are highly reliant on their social networks, but the advent of COVID-19 and the mitigation measures constrained them from accessing such networks. Also, most of the urban-slum dwellers live in overcrowded make-shift wooden kiosks (the kiosks were overcrowded with an average of about ten occupants), which are poorly ventilated. These, contributed to very difficult living conditions. Residents had to come out of their homes to access basic services and to enjoy fresh air, due to overcrowding, which led to encounters with the soldiers detailed to enforce the partial lockdown in the communities. An FGD participant intimated that while policy makers and soldiers live in good houses and could observe the COVID-19 preventive measures, they in the urban-slum live in conditions that made it difficult to comply with such measures:

*"They were in their houses [policy makers and soldiers], but we were in our kiosks. There was not a thing like, in case there is an issue, I can go to this house to eat and come out. So, the soldiers who sacked us did not try at all. We are in kiosks and we did not have a choice like when*

**Table 7. Determinants of susceptibility to the health effects of COVID-19.**

| | Afraid to seek healthcare even if unwell | Feel depressed | Feel anxious |
|---|---|---|---|
| **Age (Ref: 18–32 years)** | | | |
| 33–46 years | 0.778 | 0.890 | 0.581** |
| | (0.212) | (0.245) | (0.160) |
| 47–60 years | 0.909 | 0.887 | 0.683 |
| | (0.287) | (0.283) | (0.226) |
| 61 years and above | 0.506 | 0.450* | 0.502 |
| | (0.242) | (0.204) | (0.237) |
| **Residence (Ref: Rural)** | | | |
| Urban-slum | 0.864 | 1.839 | 3.749* |
| | (0.540) | (1.218) | (2.617) |
| **Sex (Ref: Male)** | | | |
| Female | 1.037 | 0.621** | 0.569*** |
| | (0.214) | (0.133) | (0.124) |
| **Religion (Ref: Christianity)** | | | |
| Islam | 2.563 | 1.585 | 1.012 |
| | (1.609) | (1.060) | (0.706) |
| Traditionalist | 5.552 | 0.922 | 1.297 |
| | (6.574) | (0.775) | (1.228) |
| None | 0.000000774 | 0.113 | 0.199 |
| | (0.000396) | (0.173) | (0.300) |
| **Marital status (Ref: Married)** | | | |
| Single | 0.843 | 0.536** | 0.433*** |
| | (0.215) | (0.138) | (0.113) |
| Divorced | 0.348** | 0.485 | 0.992 |
| | (0.181) | (0.242) | (0.561) |
| Widowed | 0.584 | 0.566 | 0.863 |
| | (0.293) | (0.291) | (0.443) |
| Separated | 1.168 | 0.695 | 0.841 |
| | (0.841) | (0.460) | (0.590) |
| Cohabitating | 0.528 | 0.217*** | 0.409** |
| | (0.234) | (0.0990) | (0.186) |
| **Education (Ref: None)** | | | |
| Basic | 0.681 | 0.761 | 0.971 |
| | (0.169) | (0.188) | (0.243) |
| Secondary | 0.659 | 0.767 | 0.987 |
| | (0.217) | (0.260) | (0.333) |
| Tertiary | 2.523 | 1.203 | 2.588 |
| | (1.451) | (0.716) | (1.570) |
| **Employment (Ref: Unemployed)** | | | |
| Employed | 1.345 | 1.497 | 1.826** |
| | (0.334) | (0.376) | (0.464) |
| **Income (Ref: GHC10-GHC100)** | | | |
| GHC101-GHC500 | 0.735 | 1.253 | 0.799 |
| | (0.189) | (0.329) | (0.213) |
| GHC501-GHC1000 | 0.600 | 0.643 | 0.476** |
| | (0.189) | (0.200) | (0.151) |
| GHC1001 and above | 0.324*** | 0.782 | 0.618 |

*(Continued)*

**Table 7.** (Continued)

| | Afraid to seek healthcare even if unwell | Feel depressed | Feel anxious |
|---|---|---|---|
| | (0.128) | (0.307) | (0.252) |
| Observations | 390 | 390 | 390 |
| Chi2 | 47.80 | 69.33 | 79.58 |
| P-value | 0.000453 | 0.000000234 | 4.63e-09 |

Exponentiated coefficients (odds ratios) were used; Standard errors in parentheses; * $p < 0.1$, ** $p < 0.05$, *** $p < 0.01$; Observations were less than 400 because of missing data.

*you are indoors and you are tired, you can come out to rest anywhere. So that is the problem they [soldiers] had with us, and that is why they were sacking us. We were sitting outside and that was the cause [of being beaten by the soldiers].*" (FGD, Urban-slum residents, Men under 30years).

## Coping strategies

The context in which urban-slum and rural dwellers live and the different mitigation measures introduced in these communities influenced the type of coping strategies that each category adopted.

The rural communities received support from several quarters, which facilitated the community's ability to cope with the pandemic. Thus, very few individual initiatives were reported. They reported that some of the youth from the community who were based in the urban areas and had not visited for a long time returned to the community. Other coping strategies were community members stopped saving money through community revolving loan schemes and cultivated less land due to their inability to hire labour.

The Member of Parliament for the rural communities and their environs, the district assembly, and the Ghana Water Company offered support. They provided items such as a water tank and Veronica buckets (A mechanism for hand washing originating in Ghana, which consists of a bucket of water with a tap fixed at the bottom. It is usually mounted at about two feet high with a bowl at the bottom to collect waste water) [52]. The water tank was filled with water regularly by the Ghana Water Company, to facilitate hand washing in public places. A committee comprising of members of the district assembly, district directorate of the Ghana Health Service (the service delivery body of the government's health sector) and community leaders to supervise community observance of COVID-19 preventive restrictions. Two non-governmental organizations also supported the community with nose masks and Veronica buckets. Also, the communities were taught to make locally appropriate hand washing containers and soap containers to promote hand washing in households.

Study participants in the urban-slum reported that they received little support from external sources, so they resorted to community initiatives. Nevertheless, the municipal assembly supported the community with Veronica buckets, which were placed in key areas such as outside mosques and in the market square. Also, the Ministry of Gender and Social Protection distributed cooked food to some community members. Nevertheless, study participants indicated that the distribution of cooked food was problematic, because they could not accept cooked food from unknown sources and the strategy for distributing the food was disrespectful to community and religious norms [majority of the urban-slum dwellers are Muslims].

"*The eating was an issue. We were all disturbed. We heard that people had cooked food and were sharing it to communities, but we never experienced such here. We also heard that they*

*will be cooking food to share for us. We do not know where the food that they were sharing came from. The food was placed in a Pickup truck and they threw it at people to catch. How can food be shared like that*! (FGD, Urban-slum residents, Men over 30 years)

Some community members also embarked on self-help projects such as contributing money to take care of the vulnerable especially the aged: "*We have some youth here . . ., they came out with an idea. . . We started to save one cedi* ($0.17) *each and before we realized, we had plenty rice and oil, which we distributed to all the aged. . .*" (IDI, Urban-slum resident)

Some of the urban-slum dwellers tried to escape the extreme lockdown measures by returning to their native homes in northern Ghana. They disguised themselves and tried to get onto vehicles to leave the urban area to return to their native villages in northern Ghana. However, they were arrested and sent back to the urban-slum. An FGD participant shared his experience: "*We decided to go and put on our dresses and wear another dress on them [to disguise themselves]. So that we can travel back home, but they brought us back here.*" (FGD, Urban-slum residents, men under 30 years).

Urban-slum dwellers coped with the inability to earn an income, declining economic well-being and loss of jobs by depending on relatives for remittances. Some men were able to send their wives who were pregnant and considered as vulnerable, back to their place of origin in the northern part of the country. Also, some urban-slum residents depended on their social networks for food and where one partner lost his/her job, the other supported the family.

Due to the extreme lockdown in the urban-slum, several men who were engaged in jobs within their communities such as smelting of iron could not come out to work, because the military personnel detailed to enforce the lockdown drove them away when they came out. Thus, in several of the households, women became bread winners, because they could break the compulsory lockdown, by sneaking to the market to work to earn an income.

"*Honestly, this is where my colleagues and I sit all the time. To be frank, we went through difficulties during the lockdown concerning food, but we here have friends, who are supportive. Most of us did not have anything, but we received support from these friends. The period of the lockdown was very hard, but there was support.*" (FGD, Urban-slum residents, Men over 30 years)

"*To be frank, if you are to follow all the COVID-19 preventive measures 100%* [referring to not breaking the lockdown measures], *it would be difficult taking care of your basic needs. I can say that, most of the time, our wives risk their lives to go to the market to trade. It is the females who are helping us a lot. They provide a lot of support for homes than the males.*" (FGD, Urban-slum residents, Men over 30years).

## Discussion

This study examined the socio-economic and health effects of COVID-19 among rural and urban-slum dwellers in Ghana, using a concurrent triangulation mixed methods design. Applying such a design afforded the study the opportunity to examine and understand the magnitude of the effect of COVID-19 and the mitigation measures on vulnerable populations in both urban and rural communities in a middle lower income country. The study found that COVID-19 and the government's mitigation measures had negative effects on vulnerable communities such as loss of livelihood and income, inability to get enough food for families, anxiety, depression and poor health seeking behaviour. Both rural and urban-slum dwellers adopted various coping mechanisms to deal with COVID-19 and the mitigation measures.

Majority of the study respondents confirmed the negative effects of COVID-19 on their jobs and an increase in prices of food items, which resulted in their inability to afford enough food. These findings are not surprising, since the pandemic with its associated restrictions slowed economic activities in Ghana, hence negatively affecting people's jobs. Similar findings with regard to the negative effects of COVID-19 on jobs or businesses have been reported by past studies [8,9,16,22,23]. Concerning food prices and affordability, Ghana is a major importer of food items, hence, the closure of borders may have resulted in shortage of food items, leading to a rise in their prices, making them less affordable. Similar results of price increases and food shortage were reported among refugees in Mauritania [21]. Also, in Ghana, Adom et al. [22] and Asante and Mills [24], confirmed an increase in the prices of food products due to COVID-19.

Majority of study respondents confirmed the effects of COVID-19 to be negative. Most urban-slum dwellers compared with rural dwellers, reported that they felt anxious and were depressed due to COVID-19. The findings on depression and anxiety could be attributed to the disruptions in economic activities caused by the pandemic, as well as the fear of contracting the virus. Also, it is not surprising that urban-slum dwellers complained of higher levels of anxiety and depression, considering that they experienced harsher mitigation measures compared to rural dwellers. For urban-dwellers, not having a second job such as farming to feed their families (unlike their rural counterparts), the inability to return to their native regions, being confined to their make-shift kiosks, lack of social-amenities and the inability to interact and rely more on their social networks, possibly contributed to the anxiety and the feeling of being depressed. These outcomes will have a lasting psychological effect on individuals. Khademian et al. [53], found the contrary that rural study participants reported higher levels of depression than urban participants as a result of the effects of COVID-19 and its mitigating measures. They further reported that people who lost some or all of their income during the COVID-19 outbreak experienced higher levels of stress [53]. Feter et al.'s [54] study in Brazil on the other hand reported higher rates of depressive and anxiety symptoms among women, younger age groups of between 18 to 30 years old, people diagnosed with chronic disease and people who had their income negatively affected by social restrictions.

More males reported about the negative socio-economic and health effects of COVID-19 than females. This finding could probably be due to the traditional set-up in Ghana, where men are supposed to be bread winners of households, hence, there is a relatively higher demand on them to contribute to the upkeep of the household. In the urban-slum the lockdown measures were more stringent on men, majority of whom were engaged in informal jobs within their communities. COVID-19 negatively affected their jobs, which made them felt more depressed and anxious, because of the fear of being unable to cater for the needs of their households. The women on the other hand could sneak out of the community to engage in jobs elsewhere to earn income. Nonetheless, these outcomes are not in line with Bukari et al.'s [25] study, which found that female-headed households in Ghana were more disadvantaged, as they suffered higher levels of poverty as a result of COVID-19 and it's mitigation measures. Also, Suhrcke et al. [55] reported that, relative to men, more women experienced more sleep loss problems, higher levels of unhappiness and pressure during the COVID-19 lockdown in Luxembourg. They suggested that the greater psychological difficulties that women experienced compared to men, may relate to the greater burden generally carried by women within households. Khademian et al. [53] equally found that, more female participants suffered higher levels of depression than men in Iran. Xiong et al. [56], also reported similar findings in a systematic review of China, Spain, Italy, Iran, the US, Turkey, Nepal, and Denmark.

The unwillingness of people to seek healthcare even if unwell was due to the fear of contracting the virus from health facilities, misconception about COVID-19 and a distrust of

public health facilities because of the perceived failure of health facilities to observe the standard protocols in dealing with COVID-19 cases. Study participants reported that they were afraid to access health care when they experience cough or other ailments, for it could be construed as COVID-19 or they could be tested for COVID-19 by health providers. A similar study carried out in the Cape Coast Metropolis in Ghana, found that there was a reduction in the utilisation of healthcare services as a result of the impact of COVID-19 and participants complained that they perceived negative reception at health facilities. Another study reported that study participants feared that if they visit the health facility with symptoms like cough, flu and fever, they will be treated as COVID-19 infected persons [29]. However, Saah et al. [29], reported that COVID-19 had positively impacted on the health seeking behaviour of some of their study participants. They now regularly visit health facilities for check-ups, because they have become more health conscious of the risks of chronic diseases like diabetes and hypertension and the need to manage them early.

Also, our results suggesting that younger respondents (18 to 32 years) were more likely to suffer from the socio-economic and health effects of COVID-19, could be that, younger people are yet to enter the job market, hence they depend on other people or family members for survival. Belot et al.'s [57] systematic study of China, South Korea, Japan, Italy, UK, and US corroborates the current finding. The study reported that young people experienced more drastic changes in their lives and were the most economically and psychologically affected by COVID-19. Also, a study in the UK found that, workers under the age of 25 years old felt the negative economic impact of COVID-19 more because, they were twice more likely to work in a sector that experienced a shut-down than those aged 25 years and over [58].

More urban-slum dwellers strongly agreed or were more likely to suffer from the negative socio-economic and health effects of COVID-19. This is not surprising because, the chosen urban-slum is situated in the Ashanti Region of Ghana, which is an epicenter for COVID-19, so they (urban-slum dwellers) experienced stiffer lockdown measures and military brutality. Alhassan et al.'s [59] argument that the mitigation measures adopted in Africa was 'Western elitist' mitigating measures for COVID-19, which led to excesses and extreme effects on individuals, families and economies in Africa, is apt. The lockdown in urban-slum communities was introduced without recourse to the fact that they live in overcrowded make-shift kiosks that lack social amenities. It is therefore not surprising that urban-slum residents experienced immense pain, humiliation and suffering and some flouted the lockdown restrictions. Moreover, the restrictions on movement implied that urban-slum dwellers experienced more food shortage and rising prices of food items, because, they are less agrarian compared to rural dwellers. These, may have also contributed to the higher rate of anxiety and depression among urban-slum dwellers compared to rural dwellers. Other studies also noted that poor urban-slum dwellers suffered immensely from the lockdown, which did not consider the high rate of poverty, lack of cash and social amenities in communities. Thus, they could not cope with the lockdown measures and flouted them [3,31]. In addition, Kansiime, et al. [60] noted that in Kenya and Uganda, rural farmers were less likely to experience food insecurity compared to those who depended heavily on market sources for food.

The finding that married respondents were more likely to experience the negative effect of COVID-19 on socio-economic and health indicators is surprising. This is because married individuals are expected to get financial and emotional support from their partners, to cushion them against the adverse effects of the pandemic. Notwithstanding, the outcome could be due to married people having a lot of dependents to cater for, compared to those who are single, and consequently, they may find it very difficult to afford enough food following the negative effects of the pandemic. A study in Spain on the other hand, found that married people adapted better to COVID-19 and the mitigation measures compared to the unmarried [61].

Another study in India reported that married people displayed significantly higher levels of fear compared to the unmarried and widowed [62]. However, Qian and Yahara [63] reported that unmarried, divorced and widowed people felt significantly more stress, anxiety, and depression than their married counterparts in Japan.

Low-income earners (GHC10 ($1.7) to GHC100 ($17), were more likely to suffer from the negative health effects of COVID-19, because, most low-income earners live by hand-to-mouth. Therefore, disruptions in economic activities as a result of COVID-19 may deepen their fears, leading to anxiety. This is very worrying, because it may further compound the already vulnerable condition of these low-income earners. Similarly, Martin et al. [15] found that, the lowest income earners suffered the most economically, from the effects of COVID-19 in the San Francisco Bay Area in the USA. Further, in Burkina Faso, Zidouemba et al. [64] reported that COVID-19 had resulted in food deficit among the poor and that severity of food insecurity was increasing among poor households in rural and urban areas. Moreover, non-poor rural households were likely to fall into the category of vulnerable people.

While the rural community depended on their farm produce and could access a few public amenities, which helped them to cope with the effect of COVID-19 and the mitigation measures, the case of the urban-slum residents was direr. There were very limited social amenities to rely on, thus, they relied heavily on their families and social networks for food and other basic necessities. This is not surprising, because in the Ghanaian setting, where communal living still plays an important role, it is common to rely on one's relatives, friends and neighbours in times of crises. Other studies have equally reported that social networks and interpersonal relations have helped vulnerable populations to cope with the effects of COVID-19 [65,66]. In addition, Srivastava et al. [67] reported that one of the reactions to the mitigation measures in India among internal migrants, was to return to their native homes, even though in the long-run it did not help them escape from poverty, mental health and disease.

## Conclusion and policy implications

Globally, the outbreak of COVID-19 has caused a lot of disruptions in various aspects of human life especially among vulnerable populations such as rural and urban-slum dwellers. Notwithstanding, this study found that most COVID-19 mitigating packages in Ghana did not meet the needs of vulnerable populations.

The study recommends that all vulnerable populations in urban and rural areas in Ghana should be targeted in the government's mitigating measures.

Government and its implementing agencies need to intensify community engagement in rural and urban-slums to understand community needs well in order to design appropriate intervention measures. Government should involve community representatives in designing intervention measures, since they know their communities and will be in the best position to identify the most vulnerable to bring to the attention of the government. Such an approach will contribute to effective implementation and community ownership of the mitigation measures.

In the long-term government should use the lessons from the pandemic as an incentive to intensify community development. Communities should be provided with social services and amenities, which will afford them the opportunity to enjoy basic human rights and quality life as well as facilitate resilience in the event of future pandemics.

## Study strengths and limitations

Using a concurrent triangulation mixed methods design afforded the study a comprehensive understanding of the magnitude of the problem of COVID-19 on rural and urban-slum

dwellers, as well as a thick description of the study context. The two methods complimented each other and ensured greater trustworthiness and validity of the study.

Notwithstanding the above, the present study is not without limitations. The use of perception-based measures of mental health may not be informative as clinical diagnosis. Also, comparing vulnerable and non-vulnerable populations that faced same or similar COVID-19 restrictions would have provided further insights. Moreover, using only two districts/municipalities may not be a representation of the over 200 districts/municipalities/metropolises in Ghana, hence caution must be exercised in generalising the findings to represent the entire nation. We therefore recommend future studies to take into account these issues. Nevertheless, our findings compare favourably with other studies in Ghana and in other parts of the world. Thus, the lessons learned from this study are useful for policy decision making in Ghana and beyond.

The use of convenience sampling to select portions of study respondents in the FGD could have biased the responses and compromised the study. Nevertheless, triangulating three qualitative data collection methods, enforcing quality control and triangulating the analysis and reporting process, helped to strengthen the quality of the results.

## Supporting information

**S1 Checklist.**
(DOCX)

**S1 File. IDI guide for community leaders.**
(DOCX)

**S2 File. IDI guide for community members.**
(DOCX)

**S3 File. Observation guide.**
(DOCX)

**S1 Questionnaire.**
(DOCX)

## Acknowledgments

We are grateful to the research assistants who helped in collecting data for this study, the community members and leaders of the two study sites and the government officials who granted permission and also participated in the study. We also appreciate the editor and two reviewers, whose comments and insights have helped to greatly improve the quality of this manuscript.

## Author Contributions

**Conceptualization:** Matilda Aberese-Ako, Mustapha Immurana, Maxwell Ayindenaba Dalaba, Anthony Ofosu, Margaret Gyapong.

**Data curation:** Mustapha Immurana.

**Formal analysis:** Matilda Aberese-Ako, Mustapha Immurana.

**Funding acquisition:** Fidelis E. Y. Anumu, Anthony Ofosu, Margaret Gyapong.

**Investigation:** Matilda Aberese-Ako, Mustapha Immurana, Maxwell Ayindenaba Dalaba, Fidelis E. Y. Anumu, Anthony Ofosu, Margaret Gyapong.

**Methodology:** Matilda Aberese-Ako, Mustapha Immurana, Maxwell Ayindenaba Dalaba, Anthony Ofosu, Margaret Gyapong.

**Project administration:** Matilda Aberese-Ako, Fidelis E. Y. Anumu, Margaret Gyapong.

**Resources:** Fidelis E. Y. Anumu, Anthony Ofosu, Margaret Gyapong.

**Software:** Mustapha Immurana.

**Supervision:** Matilda Aberese-Ako, Mustapha Immurana, Maxwell Ayindenaba Dalaba, Fidelis E. Y. Anumu, Anthony Ofosu, Margaret Gyapong.

**Validation:** Matilda Aberese-Ako, Mustapha Immurana, Maxwell Ayindenaba Dalaba, Anthony Ofosu, Margaret Gyapong.

**Visualization:** Matilda Aberese-Ako, Maxwell Ayindenaba Dalaba, Anthony Ofosu, Margaret Gyapong.

**Writing – original draft:** Matilda Aberese-Ako, Mustapha Immurana, Maxwell Ayindenaba Dalaba, Fidelis E. Y. Anumu.

**Writing – review & editing:** Matilda Aberese-Ako, Mustapha Immurana, Maxwell Ayindenaba Dalaba, Fidelis E. Y. Anumu, Anthony Ofosu, Margaret Gyapong.

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
