## [Decision Letter · Decision Letter 0]

25 Feb 2022

PONE-D-21-35034The socio-economic and health effects of COVID-19 among rural and urban-slum dwellers in Ghana: A mixed method approachPLOS ONE

Dear Dr. Aberese-Ako,

Thank you for submitting your manuscript to PLOS ONE. After careful consideration, we feel that it has merit but does not fully meet PLOS ONE’s publication criteria as it currently stands. Therefore, we invite you to submit a revised version of the manuscript that addresses the points raised during the review process.

We look forward to receiving your revised manuscript.

Kind regards,

Harapan Harapan, MD, PhD

Academic Editor

PLOS ONE

Journal Requirements:

2. Thank you for stating the following financial disclosure: "MG is the main recipient of funding

Financial support for this study was obtained from the World Health Organization [WHO] Alliance for Health Policy and Health Systems Research [2020/1037470-0]."

Reviewers' comments:

Reviewer's Responses to Questions

**Comments to the Author**

1. Is the manuscript technically sound, and do the data support the conclusions?

Reviewer #1: Yes

Reviewer #2: Yes

2. Has the statistical analysis been performed appropriately and rigorously? 

Reviewer #1: I Don't Know

Reviewer #2: Yes

3. Have the authors made all data underlying the findings in their manuscript fully available?

Reviewer #1: No

Reviewer #2: No

4. Is the manuscript presented in an intelligible fashion and written in standard English?

Reviewer #1: Yes

Reviewer #2: Yes

5. Review Comments to the Author

Reviewer #1: Authors have examined the socio-economic and health effects of COVID-19 among rural and urban-slum dwellers in Ghana using a mixed method approach. The study covers a very important topic that has importance in the present scenario. However, the following points have to be considered.

Major comments

-English language requires further improvement. Some sentences are difficult to understand and may require rephrasing.

-Include latitude/longitude coordinates of the study area in methods section.

-The materials and methods section should be sub-divided into sections such as Study setting, Study design and population, Sample size and sampling procedure, Data collection, Quality control and assurance, Data management and analysis, and Ethical considerations.

Minor comments

-The expansion of COVID-19 is "Coronavirus disease 2019 (COVID-19)"

-Mention all $172915600 value in millions rather than full numbers.

Reviewer #2: Thank you for the opportunity to review your manuscript on The socio-economic and health effects of COVID-19 among rural and urban-slum dwellers in Ghana: A mixed method approach. I found this article very interesting, and I offer the following feedback with the intent of strengthening the paper.

Introduction: Please kindly provide the complete picture and data regarding the prevalence of COVID-19 at rural and urban-slum dwellers in Ghana as well as the impact of COVID-19 to these settings.

Study design: Cite and elaborate any mixed methods approach that you used on this study.

Study area: You should mention the prevalence or any data of the study areas in the introduction part. Was Adaklu city also having high prevalence compare to other rural areas?

Sampling technique: Elaborate more on how to randomize your sample? Please explain on how to establish rigour and trustworthiness of mixed methods data.

Conclusion: Please make a concise, clear and short conclusion and policy implications.

Ethics statement: Give the number and year of ethics approval.

6. PLOS authors have the option to publish the peer review history of their article (what does this mean?). If published, this will include your full peer review and any attached files.

Reviewer #1: No

Reviewer #2: No

---

## [Author Response · Author response to Decision Letter 0]

12 May 2022

Response to Editor’s comments 

Response: Authors have ensured that the manuscript has met the PLOS ONE requirements, including those for file naming

2. Please state what role the funders took in the study. If the funders had no role, please state 

Response: The funders had no role in study design, data collection and analysis, decision to publish, or preparation of the manuscript.

3. We note that you have indicated that data from this study are available upon request. PLOS only allows data to be available upon request if there are legal or ethical restrictions on sharing data publicly. For more information on unacceptable data access restrictions, please see http://journals.plos.org/plosone/s/data-availability#loc-unacceptable-data-access-restrictions. In your revised cover letter, please address the following prompts:

a. If there are ethical or legal restrictions on sharing a de-identified data set, please explain them in detail (e.g., data contain potentially sensitive information, data are owned by a third-party organization, etc.) and who has imposed them (e.g., an ethics committee). Please also provide contact information for a data access committee, ethics committee, or other institutional body to which data requests may be sent. 

Response: There are ethical restrictions on sharing the data.

The address is the Administrator, Research Ethics Committee University of Health and Allied Sciences, PMB 31, Ho, Volta region. Email: rec@uhas.edu.gh

b. If there are no restrictions, please upload the minimal anonymized data set necessary to replicate your study findings as either Supporting Information files or to a stable, public repository and provide us with the relevant URLs, DOIs, or accession numbers. For a list of acceptable repositories, please see http://journals.plos.org/plosone/s/data-availability#loc-recommended-repositories. We will update your Data Availability statement on your behalf to reflect the information you provide.

Response: There are ethical restrictions

Response: Thank you 

4. Your ethics statement should only appear in the Methods section of your manuscript. If your ethics statement is written in any section besides the Methods, please move it to the Methods section and delete it from any other section

Response: Ethics statement included in methods section, please see lines 254-263

5. Please include captions for your Supporting Information files at the end of your manuscript, and update any in-text citations to match accordingly.

Response: Captions for supporting information files included at the end of the manuscript, please see lines 792 -797

- Reviewer #1:

Thank you very much, for your review. Please, find below the responses to your comments

Major comments

-English language requires further improvement. Some sentences are difficult to understand and may require rephrasing.

Response: English language improved in all sections of the manuscript. 

-Include latitude/longitude coordinates of the study area in methods section.

Response: latitude/longitude coordinates of the study area in methods section, kindly see lines 132 and 133 and 139

-The materials and methods section should be sub-divided into sections such as Study setting, Study design and population, Sample size and sampling procedure, Data collection, Quality control and assurance, Data management and analysis, and Ethical considerations.

Response: Study setting, Study design and population, Sample size and sampling procedure, Data collection, Quality control and assurance, Data management and analysis, included for quantitative and qualitative methods. Kindly see lines 123 -253

Response” Ethical considerations and ethics approval number included, kindly see lines 255-263

Minor comments

-The expansion of COVID-19 is "Coronavirus disease 2019 (COVID-19)"

Response: Coronavirus disease 2019 (COVID-19) corrected. Kindly see line 77

-Mention all $172915600 value in millions rather than full numbers.

Response: Value changed to millions rather than full numbers. Kindly see lines 92 and 94

Reviewer #2: Thank you for your insight, very useful comments and commendation.

Response to comments

Introduction: Please kindly provide the complete picture and data regarding the prevalence of COVID-19 at rural and urban-slum dwellers in Ghana as well as the impact of COVID-19 to these settings.

Response: The prevalence of COVID-19 in the study regions have been reported, kindly see lines see 89-91. COVID-19 cases are reported by administrative regions, Ghana does not record prevalence rates by rural and urban settlements, so we are unable to provide such data. 

Response: The impact of COVID-19 on rural and urban-slum dwellers in Ghana reported. Kindly see lines 108-116.

Study design: Cite and elaborate any mixed methods approach that you used on this study.

Response: Mixed methods approach cited and elaborated, kindly see lines 145-153

Study area: You should mention the prevalence or any data of the study areas in the introduction part. Was Adaklu city also having high prevalence compared to other rural areas?

Response: The prevalence or any data of the study regions have been reported, kindly see lines 90 & 91. The reason for the inclusion of Adaklu district in the study has been explained, kindly see lines 134-142.

Sampling technique: Elaborate more on how to randomize your sample

Response: Elaboration of randomized sample explained, kindly see lines 166-169.

Please explain on how to establish rigour and trustworthiness of mixed methods data.

Response: Rigour and trustworthiness of the study have been reported in the methodology section. Also, the GRAMMS checklist was used and has been included as Supporting Information. Kindly see lines 151 and 152

Conclusion: Please make a concise, clear and short conclusion and policy implications.

Response: a concise, clear and short conclusion and policy implications has been included, kindly see lines 727-755

Ethics statement: Give the number and year of ethics approval.

Response: Please, the number and year of ethics approval have been reported in lines 255-256.

---

## [Decision Letter · Decision Letter 1]

4 Jul 2022

The socio-economic and health effects of COVID-19 among rural and urban-slum dwellers in Ghana: A mixed methods approach

PONE-D-21-35034R1

Dear Dr. Aberese-Ako,

We’re pleased to inform you that your manuscript has been judged scientifically suitable for publication and will be formally accepted for publication once it meets all outstanding technical requirements.

Kind regards,

Harapan Harapan, MD, PhD

Academic Editor

PLOS ONE

Additional Editor Comments (optional):

Reviewers' comments:

Reviewer's Responses to Questions

**Comments to the Author**

1. If the authors have adequately addressed your comments raised in a previous round of review and you feel that this manuscript is now acceptable for publication, you may indicate that here to bypass the “Comments to the Author” section, enter your conflict of interest statement in the “Confidential to Editor” section, and submit your "Accept" recommendation.

Reviewer #1: All comments have been addressed

2. Is the manuscript technically sound, and do the data support the conclusions?

Reviewer #1: Yes

3. Has the statistical analysis been performed appropriately and rigorously? 

Reviewer #1: Yes

4. Have the authors made all data underlying the findings in their manuscript fully available?

Reviewer #1: Yes

5. Is the manuscript presented in an intelligible fashion and written in standard English?

Reviewer #1: Yes

6. Review Comments to the Author

Reviewer #1: Thank you for considering my comments while revising the manuscript. The following minor correction should be made.

M&M - Study settings should contain the description of the settings used in the study. Therefore shift the description on the rationale behind the selection of that particular region to the "discussion" part.

7. PLOS authors have the option to publish the peer review history of their article (what does this mean?). If published, this will include your full peer review and any attached files.

Reviewer #1: No

---

## [Editor Report · Acceptance letter]

8 Jul 2022

PONE-D-21-35034R1 

The socio-economic and health effects of COVID-19 among rural and urban-slum dwellers in Ghana: A mixed methods approach 

Dear Dr. Aberese-Ako:

I'm pleased to inform you that your manuscript has been deemed suitable for publication in PLOS ONE. Congratulations! Your manuscript is now with our production department. 

Kind regards, 

on behalf of

Dr. Harapan Harapan 

Academic Editor

PLOS ONE